



# A Grid Model for Vertical Correction of Precipitable Water Vapor over the Chinese Mainland and Surrounding Areas Using Random Forest

Junyu Li[1,2], Yuxin Wang[1,2], Lilong Liu[1], Yibin Yao[3], Liangke Huang[1], Feijuan Li[1,2]

[1] College of Geomatics and Geoinformation, Guilin University of Technology, Guilin, China.
[2] Guangxi Key Laboratory of Spatial Information and Geomatics, Guilin, China.
[3] School of Geodesy and Geomatics, Wuhan University, Wuhan, China.

*Correspondence to*: Junyu Li (junyu_li@whu.edu.cn)

**Abstract.** Various ground-based observing techniques provide precipitable water vapor (PWV) products with different

spatial resolutions. To effectively integrate these products, especially in terms of vertical orientation, spatial interpolation is essential. In this context, we have developed a model to characterize PWV variation with altitude in the study area. Our model, known as RF-PWV (a PWV vertical correction grid model with a 1° x 1° resolution), is constructed using random forest based on the relationship between PWV differences from the fifth-generation European Centre for Medium-Range Weather Forecasts reanalysis (ERA5) monthly average hourly data and height differences and time. When validated against

1-h ERA5 PWV profiles, RF-PWV exhibits a 99.84% reduction in Bias and a 63.41% decrease in RMSE compared to the most recent model, C-PWVC1. Furthermore, when validated against radiosonde data, RF-PWV shows a 96.36% reduction in Bias and a 5% decrease in RMSE compared to C-PWVC1. Additionally, RF-PWV outperforms C-PWVC1 in terms of resistance to seasonal and height differences interference. The model eliminates the need for meteorological parameters, allowing for high-precision PWV vertical correction by inputting only time and height differences. Consequently, RF-PWV

can significantly reduce errors in vertical correction, enhance PWV fusion product accuracy, and provide insights into PWV vertical distribution, thereby contributing to climate research.

## 1 Introduction

Precipitable water vapor (PWV), the most abundant greenhouse gas, primarily resides in the troposphere and plays a pivotal role in the global energy budget, hydrological cycle, and climate change (Zhang et al., 2018; Li et al., 2022b; Dessler and

Sherwood, 2009; Raval and Ramanathan, 1989; Rocken et al., 1997). Various observation platforms, including radiosondes (RS), microwave water vapor radiometers (WVR), satellite remote sensing, ground-based global navigation satellite systems (GNSS), and reanalysis data, have amassed extensive PWV data through long-term data accumulation (Huang et al., 2022). Combining multi-source data enables more accurate and comprehensive water vapor monitoring and meteorological research (Zhang et al., 2019a; Li et al., 2022a; Alshawaf et al., 2015; Lindenbergh et al., 2009). However, inconsistent pressure levels

(heights) for storing PWV data from different sources hinder the fusion and reliability analysis of PWV multi-source data. Therefore, precise PWV vertical corrections are indispensable for the utilization of PWV fusion products. Additionally,





PWV vertical correction is essential for obtaining PWV's vertical distribution characteristics, which are crucial for weather forecasting and climate research. Hence, proposing a more accurate and applicable PWV vertical correction model is of paramount importance.

Common methods for PWV vertical correction involve establishing empirical vertical correction models to enhance the applicability of PWV vertical correction (Emardson and Johansson, 1998; Dousa and Elias, 2014; Huang et al., 2023).Reitan (1963) introduced an empirical formula describing water vapor density's exponential decrease in the vertical direction, based on the relationship between PWV near the surface and at high altitudes. The PWV lapse rate (-0.5 mm/km), estimated by Kouba (2008) using the International GNSS Service (IGS) and the Vienna Mapping Function 1 (VMF1), has been widely

adopted. However, considering the seasonal variations of the PWV lapse rate as constant introduces significant errors in PWV vertical correction (Tomasi, 1977; Leckner, 1978; Zhang et al., 2019b; Zhang et al., 2022). Huang et al. (2021) developed a PWV vertical correction model that accounts for seasonal variations in the PWV lapse rate, offering greater accuracy and stability than the classical PWV vertical correction model (PWV lapse rate = –0.5 mm/km) in China. Wang et al. (2022) incorporated spherical harmonic functions to develop a PWV vertical correction model, achieving high accuracy

in the Qinghai-Tibetan Plateau. Nevertheless, many existing models assume PWV's exponential decrease and represent PWV lapse rate variations using periodic functions, failing to address complex nonlinear variations beyond daily/sub-daily and seasonal variations of the PWV lapse rate.

Neural network techniques are well-suited for handling nonlinear problems and have found applications in various industries (Zheng et al., 2022). Machine learning has demonstrated promising potential in modeling tropospheric parameters (Ravuri et

al., 2021; Lam et al., 2022). Senkal (2015) developed a model for predicting PWV in Turkey using a Resilient Propagation (RP) neural network, which provides PWV estimates for a given location. Validation with RS PWV data in the study area revealed good agreement between the new model and RS PWV data. Zhu et al. (2022) created a weighted mean temperature (Tm) vertical correction grid model (CTm-FNN) employing a feedforward neural network in China. This model outperformed the Chinese Tropospheric Model (CTrop) and Global Pressure and Temperature 3 (GPT3), reducing RMSE by

86% and 83%, respectively.

Therefore, this paper presents a vertical correction grid model (RF-PWV) for China and surrounding areas, harnessing Random Forest's powerful nonlinear fitting capability and the high temporal resolution of monthly average hourly PWV data. With RF-PWV, PWV differences can be obtained by simply inputting time and height differences, allowing for high-precision PWV vertical correction. The model offers PWV vertical correction techniques for multi-source PWV fusion,

weather forecasting, and climate studies.

We begin by providing an overview of the study area and the experimental dataset. Subsequently, we describe the data processing strategy and modeling methodology. Next, we evaluate the performance of the RF-PWV model. Finally, we conclude our study and outline future directions.



## 2 Data and methods

### 2.1 Study area

The study area includes the region between 15°N and 55°N latitude and 70°E to 135°E longitude, covering mainland China and its surrounding areas, characterized by extensive land and ocean. China's topography exhibits significant variation, with higher elevations in the west gradually sloping to lower elevations in the east. Influenced by the monsoon climate, the summer monsoon brings substantial moisture from the ocean into the region, while winter introduces cold, dry air inland

(Sun et al., 2019; Zhang et al., 2019c). These geographical and climatic factors contribute to a complex spatiotemporal variation in PWV. As a result, the vertical distribution of PWV in this area presents a challenging problem to characterize, making it a suitable choice for our experimental area.

### 2.2 Datasets

### 2.2.1 ERA5 PWV

ERA5, the fifth-generation atmospheric reanalysis product developed by the European Centre for Medium-Range Weather Forecasts (ECMWF), offers access to 1-h meteorological data across 37 pressure levels, with a horizontal resolution as fine as 0.25° x 0.25°. This dataset can be downloaded from https://cds.climate.copernicus.eu/ (Albergel et al., 2018). ERA5 is renowned for its superior accuracy compared to its predecessor, ERA-Interim, and has gained widespread usage in meteorological research (Hersbach et al., 2020; Lu et al., 2023; Chen et al., 2023). Moreover, the monthly averaged dataset,

in terms of accuracy, rivals the daily dataset while demonstrating greater stability (Dogan and Erdogan, 2022). Additionally, the monthly average hourly dataset offers the advantage of capturing both seasonal variations in meteorological data and finer-grained sub-daily variations. In this study, we utilize the monthly average hourly dataset, which provides 1-h data at 37 pressure levels with a spatial resolution of 1° x 1°. The PWV for each pressure level is determined through integration, as described by (Zhang et al., 2019d; Wang et al., 2016):

$$PWV = \sum_{i}^{n-1} \frac{(q_i + q_{i+1}) \bullet (p_{i+1} - p_i)}{2 \bullet \rho_w \bullet g}, \tag{1}$$

$$g = 9.780325 \bullet \left[\frac{1 + 0.00193185 \bullet \sin(\varphi)^2}{1 - 0.00669435 \bullet \sin(\varphi)^2}\right]^{0.5}, \tag{2}$$

where $n$ represents the total number of layers, $q_i$ and $p_i$ represent the specific humidity (kg/kg) and pressure (Pa) at the $i$ layer, respectively; $\rho_w$ is the density of liquid water, which is standardized to 1,000 kg/m3; $g$ is the gravitational acceleration (m/s), $\varphi$ denotes the latitude (rad).

It is crucial to emphasize that the upper boundary of the troposphere lies at approximately 10 km altitude (Ding, 2020). Consequently, PWV effectively approaches 0 mm when situated at elevations exceeding 12 km vertically. As a result, we restrict our PWV calculations to cover pressure levels within the range of 0 to 12 km above the grid point for all subsequent analyses and investigations.



### 2.2.2 RS PWV

RS are widely recognized for their high-precision PWV measurements and are commonly considered a reference standard for evaluating other measurement techniques (Adeyemi and Joerg, 2012; Wang et al., 2021; Zhao et al., 2022). We obtained RS PWV data from the Integrated Global Radiosonde Archive (IGRA), accessible at https://www1.ncdc.noaa.gov/pub/data/IGRA, with a temporal resolution of 12 h. We made use of meteorological data from 148 stations, focusing on pressure levels within the 0 - 12 km altitude range (as illustrated in Figure 1). The specific

humidity at each pressure level was determined by employing Eq (3) and (4), which are as follows (Zhai and Eskridge, 1997; Ross and Elliott, 1996):

$$e = \frac{RH \bullet e_s}{100}, \tag{3}$$

$$q = \frac{0.622 \bullet e}{p - 0.378 \bullet e}, \tag{4}$$

where $RH$ represents relative humidity (%), $e_s$ signifies saturated vapor pressure (Pa), $e$ denotes water vapor pressure (Pa),

and $q$ represents specific humidity (kg/kg). Subsequently, RS PWV values for various pressure levels were calculated using Eq (1).

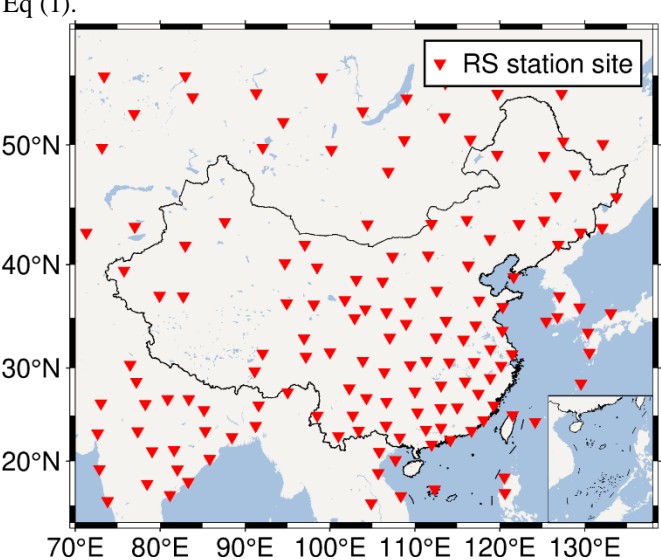

**Figure 1. Distribution of the selected radiosonde sites.**

### 2.3 Establishment of the RF-PWV model

The Random Forest is an ensemble learning method that uses multiple decision trees, initially introduced by Breiman and Cutler in 2001 (Breiman, 2001). It operates by constructing decision trees during the training process and subsequently averaging the results from all these trees. This method offers the advantage of rapid training and the capability to handle



intricate nonlinear relationships between input and output variables. The equation governing Random Forest's prediction is expressed as follows:

$Y(X) = \frac{1}{B}\sum_{b=1}^{B} T_b(X),$                                    (5)

where $Y(X)$ is the final prediction result, $T_b(X)$ represents the predicted value of each decision tree, and $B$ denotes the number of decision trees. The selection of an appropriate number of decision trees is pivotal in modeling; too few trees may lead to overfitting, while too many trees can result in excessively long modeling times (Sun et al., 2021; Probst and Boulesteix, 2017).

### 2.3.1 Defining the primary parameter

To assess the performance of machine learning models, 10-fold cross-validation is a commonly employed technique (Rodriguez et al., 2010; Zhang and Yao, 2021). In this context, 10-fold cross-validation was employed to ascertain the optimal number of decision trees based on Root Mean Square Error (RMSE). The fundamental principle of 10-fold cross-validation entails dividing the input data into ten groups. Subsequently, nine randomly selected groups are utilized as the training set, and the remaining group serves as the test set. This process is reiterated ten times to ensure that all data is included in both training and testing. This approach provides results that closely approximate the accuracy of the final model while guarding against overfitting (Santos et al., 2018). Based on our experience, we experimented with decision tree numbers ranging from 5 to 95, with a step size of 10, to train the model and evaluate its performance under varying decision tree quantities (Li et al., 2023). The results, depicted in Figure 2, exhibit a significant decline in RMSE as the number of decision trees increases from 5 to 75, reaching a minimum at 75. However, increasing the number of trees to 75 does not significantly enhance accuracy, and it incurs longer training times. In consideration of the need for modeling at multiple grid points and balancing fitting quality with training time, a final decision was made to employ 55 trees for building the model.

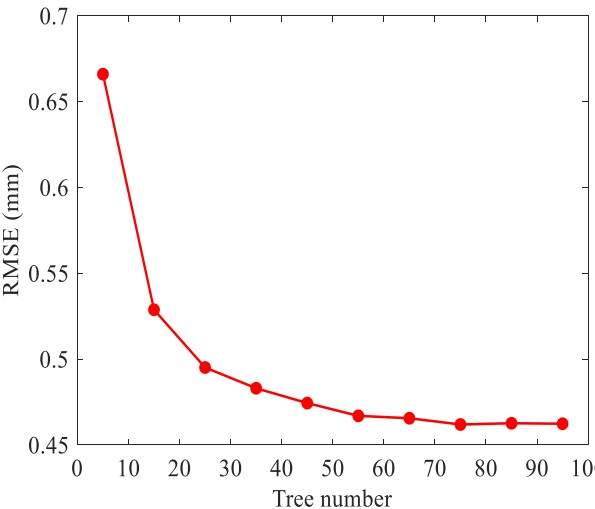



**Figure 2. Cross-validation RMSE in different numbers of decision trees**

### 2.3.2 Training the model

During the model training phase, we performed individual modeling at each grid point ($1° \times 1°$) using ERA5 monthly average hourly PWV data at pressure levels ranging from 1000 to 225 hPa, within the 0 - 12 km altitude range, spanning the years 2008 to 2017. The PWV differences ($\Delta PWV$) and height differences ($\Delta GPH$) for each pressure level relative to the bottom level were computed and utilized as the training dataset. In essence, each grid point contained 63,360 samples ($22 \times 24 \times 12 \times 10$), and the region consisted of 2,706 grid points ($66 \times 41$) at $1° \times 1°$ resolution. The model, denoted as the RF-PWV model, characterizes the relationship between $\Delta PWV$ and $\Delta GPH$, as illustrated in Figure 3. The input data included year, day of the year (doy), hour of the day (hod), and $\Delta GPH$; the output data were $\Delta PWV$ When users employ the model, they are only required to provide the geopotential height of the target point, the reference PWV, the time (year, doy, hod), and the height difference of the target point concerning the datum point ($\Delta GPH$). Then, the user can obtain the corresponding $\Delta PWV$ and add reference PWV to the $\Delta PWV$ to get the PWV of the target height.

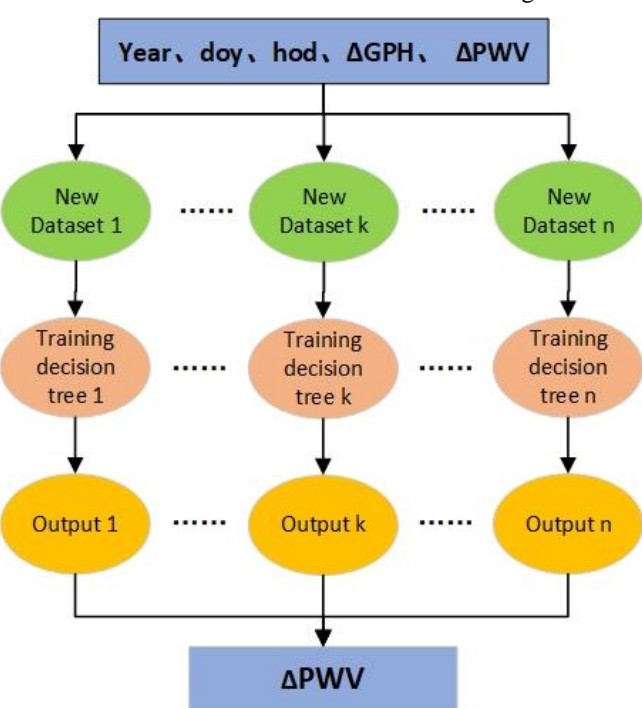

**Figure 3. Network structure of RF-PWV model based on random forest algorithm**

In the application of the RF-PWV model, the four grid points surrounding the target point are determined based on the target point's geographical coordinates (latitude and longitude). Subsequently, the target point. Then, the $\Delta PWV$ at the corresponding height of the four selected points is calculated using the RF-PWV model. Finally, the $\Delta PWV$ at the target point's location is determined through bilinear interpolation. This process involves calculating the difference between the



target point's $GPH$ and the reference station's $GPH_0$ to get the $\Delta GPH$. Next, the time information is input into the models for the four nearest grid points to the target point, yielding the $\Delta PWV$ at the corresponding height of these grid points. Finally, bilinear interpolation is employed to calculate the $\Delta PWV$ at the target point's location. This method offers the advantage of not requiring an exceptionally strong spatial generalization ability for a single model. It comprehensively considers the relationship between the target point and the four nearest grid points within the limited spatial context, resulting in enhanced consistency and higher accuracy at each grid point, ensuring the overall model's robustness.

### 3 Accuracy validation and analysis

To validate the RF-PWV model, we employed hourly ERA5 and RS pressure level data from the study area in 2018, while also selecting a newly developed PWV vertical correction model (C-PWVC1) for comparison. The accuracy metrics employed for evaluation are Bias and RMSE, as outlined below:

$$Bias = \frac{1}{n}\sum_{i=1}^{n}\left(X_i - X_i^{'}\right), \tag{6}$$

$$RMSE = \sqrt{\frac{1}{n}\sum_{i=1}^{n}(X_i - X_i^{'})^2}, \tag{7}$$

where $X^{'}$ is the reference values, $X$ denotes model outputs, and $n$ is the number of samples.

### 3.1 Validation of RF-PWV using ERA5 PWV

The RF-PWV model and C-PWVC1 were applied to vertically correct the hourly ERA5 bottom-level PWV data (1°×1°) for the year 2018 to other pressure levels within the 0 - 12 km altitude range, excluding the bottom level. The results were then compared with ERA5 data, and the overall Bias and RMSE are presented in Table 1. RF-PWV exhibited a Bias close to 0 mm, indicating minimal systematic Bias between the interpolated PWV and ERA5 PWV. Moreover, it reduced Bias by 1.42 mm compared to C-PWVC1, corresponding to a remarkable optimization of 99.84%. The Bias values for RF-PWV were observed to fluctuate slightly within the range of –0.01 to 0.01 mm. Additionally, the RF-PWV RMSE showed a substantial reduction of 63.40% compared to C-PWVC1. Furthermore, the RMSE values for RF-PWV demonstrated a more stable fluctuation pattern with a considerably narrower range. Overall, RF-PWV exhibited significantly higher accuracy than C-PWVC1, with corrected results showing better agreement with the reference values.

**Table 1 Validation results of the RF-PWV and C-PWVC1 models tested by ERA5 data**

| Model | Bias (mm) | | | RMSE (mm) | | |
|---|---|---|---|---|---|---|
| | Mean | Min | Max | Mean | Min | Max |
| RF-PWV | 0.00 | -0.01 | 0.01 | 0.75 | 0.39 | 1.22 |
| C-PWVC1 | 1.42 | -0.96 | 3.65 | 2.05 | 0.72 | 4.25 |



To provide a spatial illustration of the models' accuracy consistency, Figure 4 displays the Bias and RMSE values for each grid point for both RF-PWV and C-PWVC1. Notably, C-PWVC1 exhibited a significant north-south difference in Bias, with larger values in the south and smaller values in the north. Most areas displayed a positive Bias, except for a pronounced negative Bias in the Qinghai-Tibetan Plateau. In contrast, RF-PWV demonstrated a substantial reduction in Bias across

almost all grid points in the study area, approaching 0 mm, effectively eliminating the north-south discrepancy. The most noteworthy improvement in accuracy was observed in the Qinghai-Tibet Plateau and low-latitude regions. Despite the challenging climate conditions in the Qinghai-Tibet Plateau and the strong land-sea interactions in the study area's low latitudes, which contribute to complex PWV variations, RF-PWV still achieved a Bias close to 0 mm. These results highlight RF-PWV's adaptability to diverse weather conditions and its wide applicability. Furthermore, C-PWVC1 displayed a north-

south difference in RMSE, with values gradually decreasing from south to north. Higher RMSE values were concentrated in the southwestern and southeastern regions, reaching a maximum of 4.25 mm. This phenomenon is mainly attributable to the proximity of these regions to the ocean, frequent water vapor exchange between land and sea, and the complexity of PWV variations. However, RF-PWV's RMSE in these regions was significantly smaller than that of C-PWVC1, consistently measuring below 2 mm. Overall, RF-PWV's RMSE was lower than that of C-PWVC1 across the study area. Furthermore,

RF-PWV exhibited excellent agreement, with values mostly hovering around 0.75 mm, nearly independent of spatial variations. These outcomes underscore the higher accuracy and improved spatial accuracy consistency of RF-PWV across the study area.



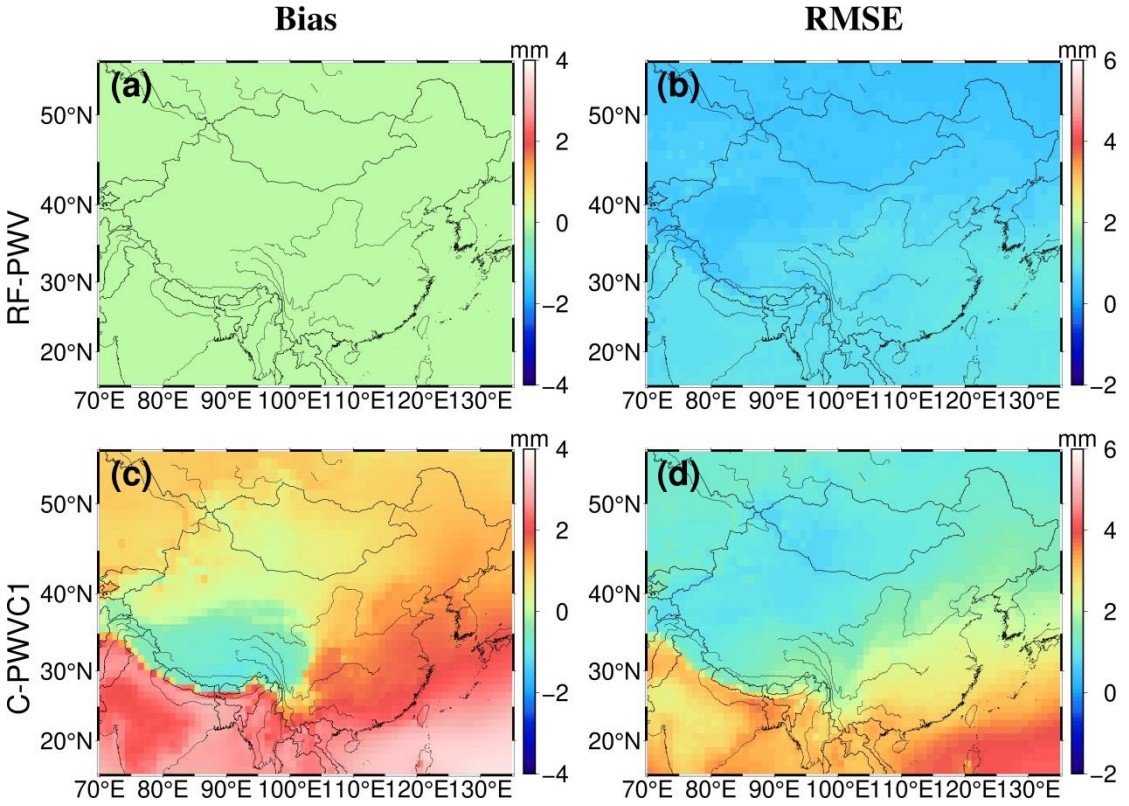

**Figure 4. Distributions of Bias and RMSE for the RF-PWV and C-PWVC1 with respect to the ERA5 data**

To further evaluate the models' performance across different seasons, we calculated the Bias and RMSE values for four representative grid points using data from 2018. These grid points were selected to represent various regions: (80.00°E, 40.00°N) in the northwestern region, (95.00°E, 15.00°N) in the southwestern region, (110.00°E, 25.00°N) in the southeastern region, and (125.00°E, 45.00°N) in the northeastern region. Figures 5a, 5b, 5g, and 5h illustrate that C-PWVC1 exhibited the highest Bias and RMSE values during June-September, reaching 5.41 mm and 6.23 mm at (80.00°E, 40.00°N) and 6.85 mm

and 7.75 mm at (125.00°E, 45.00°N), respectively. Conversely, the lowest Bias and RMSE values were recorded during January-February and November-December, hovering around 0 mm, with discernible seasonal fluctuations. This pattern is primarily attributed to significant PWV variations during the wet and rainy northern summers, contrasted with relatively mild PWV variations during the cold and dry winters. In contrast, Figures 5c, 5d, 5e, and 5f show that the seasonal differences in Bias and RMSE for C-PWVC1 were less pronounced in the southern regions than in the northern regions. At

(110.00°E, 25.00°N), which experiences abundant PWV changes and heavy rainfall throughout the year, the model's accuracy was relatively lower, with no noticeable seasonal variations. Similarly, near the equator (95.00°E, 15.00°N), overall Bias and RMSE values were more significant, with minimal seasonal differences. RF-PWV exhibited seasonal variations characterized by lower accuracy in summer and higher accuracy in winter across the northern study area, consistent with C-PWVC1 but with smaller variations. Notably, RF-PWV achieved substantially lower Bias and RMSE values than C-PWVC1





during the summer months. Figures 5a, 5b, 5g, and 5h demonstrate that RF-PWV effectively reduced Bias and RMSE at grid points in the northern region (80.00°E, 40.00°N; 125.00°E, 45.00°N), with Bias reductions of 98.84% and 99.10% and RMSE reductions of 58.47% and 72.99%, respectively. Throughout the year, RF-PWV's Bias and RMSE exhibited relatively stable patterns, with minimal fluctuations around 0 mm. Conversely, Figures 5c, 5d, 5e, and 5f reveal that RF-PWV maintained Bias and RMSE values around 0 mm, offering greater accuracy compared to C-PWVC1 in the southern grid

points. In summary, RF-PWV exhibited enhanced resistance to seasonal variations, maintaining stable and accurate performance throughout the year across the study area.

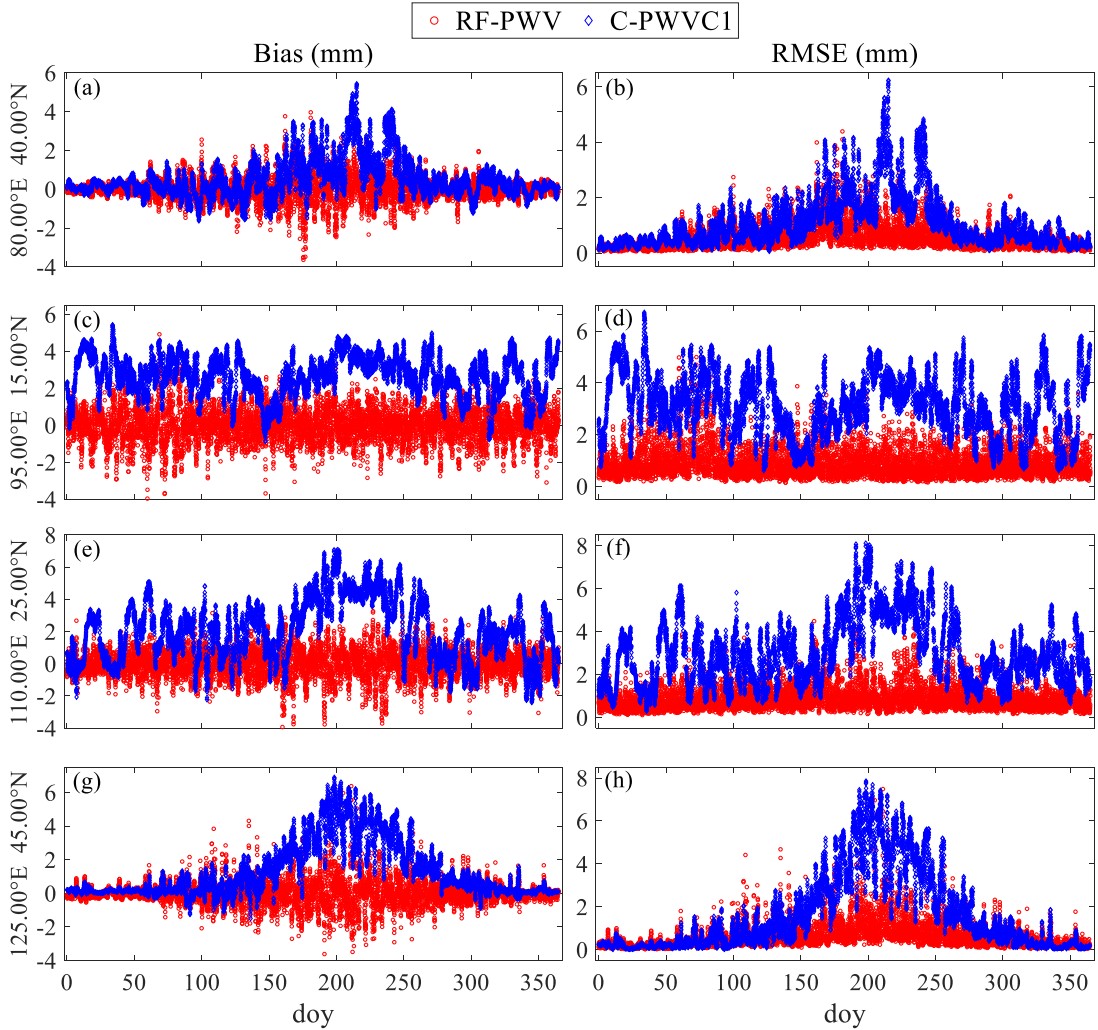

**Figure 5. Time series of RF-PWV and C-PWVC1 Bias and RMSE on four selected grid points**

Given that more than three-quarters of the water vapor is concentrated in the lower atmosphere, in practice, most of the

vertical correction of PWV occurs in the lower atmosphere (Yang et al., 2020). Bias and RMSE for C-PWVC1 and RF-PWV are statistically determined based on height differences, divided into 12 sections ranging from 0 to 6 km with intervals

of 0.5 km. This division helps assess the applicability of the two models across different height segments. The results are presented in Figure 6. Notably, C-PWVC1 exhibits a positive Bias in every height difference segment, with the Bias increasing as the height difference rises from 0 to 2.5 km, ultimately stabilizing at around 2.0 mm. RF-PWV Bias tends to

approach 0 mm within the height difference of 0 to 2.5 km but shows a negative Bias beyond this range, with the absolute value increasing and reaching a maximum value of less than 0.2 mm. In each height difference segment between 0 to 6 km, RF-PWV Bias is closer to 0 mm than C-PWVC1 Bias, indicating that the corrected value of RF-PWV is more consistent with the reference value across different height difference segments. Additionally, RF-PWV RMSE is significantly smaller than C-PWVC1 in all height difference segments. The RMSE for C-PWVC1 exhibits the same increasing trend as Bias,

stabilizing around 3 mm after the height difference exceeds 2.5 km. In contrast, the RF-PWV RMSE is less than 1 mm in all height difference segments. These findings demonstrate that RF-PWV offers improved correction effectiveness and higher accuracy compared to C-PWVC1. Consequently, RF-PWV exhibits superior performance and greater accuracy consistency within each height difference segment, indicating that it is less influenced by variations in height difference. This enhanced adaptability to height differences enables a finer-scale description of the vertical distribution of PWV.

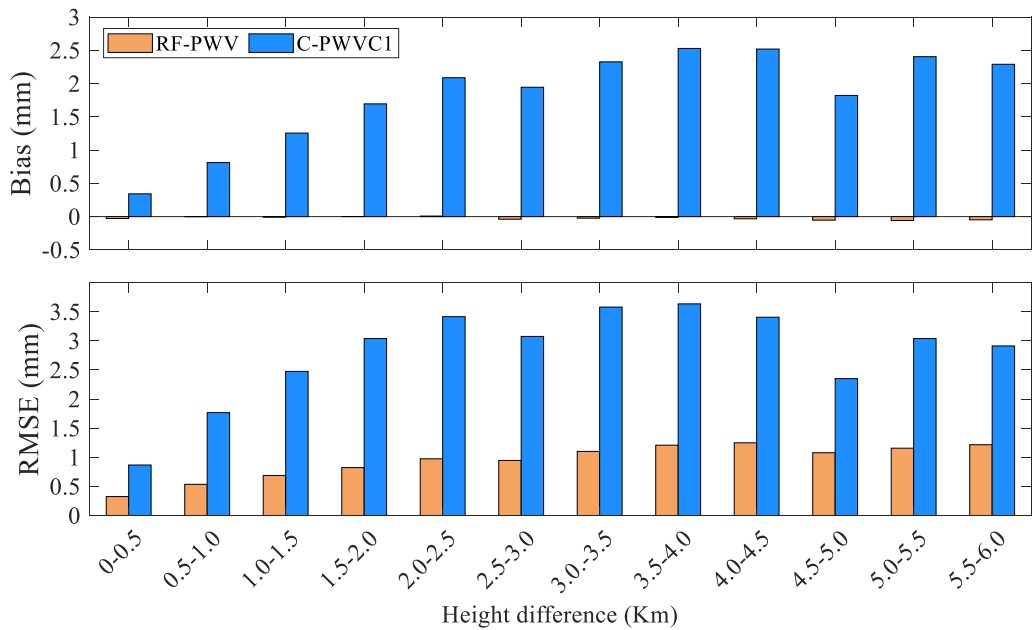


**Figure 6. Accuracy of RF-PWV and C-PWVC1 in each height difference with respect to ERA5 data.**

### 3.2 Validation of RF-PWV using RS PWV

To further validate the applicability of RF-PWV, the PWV data for all pressure levels within the 0–12 km altitude range from 148 RS stations in 2018 were used to assess the accuracy of RF-PWV and C-PWVC1. Since the sounding stratified

data are not uniformly distributed vertically, the variation of PWV with elevation was fitted using an exponential function





based on the 2018 PWV data from each sounding station. Using the fitting results, the PWVs of neighboring levels were interpolated using inverse distance weighting (IDW) to generate a sequence of PWVs within the range of 0–12 km with intervals of 500 m. This sequence of PWVs served as reference values. For each RS station, the four grid points (1° × 1°) in proximity were selected, and the output   of these four grid point models was bilinearly interpolated to the RS station to

obtain the RF-PWV result. To account for systematic Bias between modeling data and RS data, the average difference between the corrected RF-PWV value and the corresponding original value was computed as the systematic bias for each level at the RS station. Finally, the statistical accuracy of RF-PWV and C-PWVC1, after eliminating the systematic bias, is presented in Table 2.

**Table 2. Validation results of the RF-PWV and C-PWVC1 models tested by RS data**

| Model | Bias (mm) | | | RMSE (mm) | | |
|---|---|---|---|---|---|---|
| | Mean | Min | Max | Mean | Min | Max |
| RF-PWV | 0.05 | -0.25 | 0.33 | 2.59 | 0.94 | 4.89 |
| C-PWVC1 | 1.36 | -6.62 | 3.46 | 2.71 | 0.72 | 16.55 |

Table 2 reveals that the accuracy of C-PWVC1 is significantly lower than that of RF-PWV. C-PWVC1 exhibits a Bias of 1.36 mm, ranging from –6.62 to 3.46 mm, whereas RF-PWV Bias is only 0.05 mm, reduced by 1.31 mm and improved by 96.36%. The range of variation is notably reduced to –0.25 to 0.33 mm. Moreover, RF-PWV RMSE is considerably smaller and more stable, with RMSE reduced to 2.59 mm, ranging from 0.49 to 4.89 mm, corresponding to a decline rate of approximately 5% compared to C-PWVC1. Consequently, RF-PWV demonstrates superior accuracy and stability in vertical

PWV correction at 148 RS stations in the study area.

The Bias and RMSE for each RS station are also computed to further illustrate the application capabilities of the two models, as shown in Figure 7. As depicted in Figure 7a and b, C-PWVC1 exhibits a positive Bias on almost all stations except for the RS stations in the Yunnan-Guizhou Plateau, where the Bias is less pronounced. In contrast, RF-PWV Bias is consistently less than 0.5 mm and closer to 0 mm. Compared to C-PWVC1, the absolute value of RF-PWV Bias is effectively reduced in

the Yunnan-Guizhou Plateau region, with the most significant reduction reaching 3.13 mm. Meanwhile, positive Bias in other areas is also reduced to varying degrees. Figure 7b and d demonstrate that RF-PWV RMSE exhibits a certain degree of reduction compared to C-PWVC1, with the most substantial decline occurring in the sites located in the Yunnan-Guizhou Plateau. In this region, the corresponding RMSE for C-PWVC1 is consistently larger than 8 mm, with a maximum value of 16.54 mm. In contrast, RF-PWV RMSE at all RS stations is less than 5 mm, with a maximum RMSE reduction of 11.65 mm.

Given the complex terrain and significant undulations in the Yunnan-Guizhou Plateau, where the difference in height between the target point and the reference grid can be up to 1–2 km (Chen et al., 2011). Therefore, RF-PWV demonstrates superior performance and more stable accuracy compared to C-PWVC1 across the entire study area. This advantage is particularly pronounced in regions with significant variations in height.



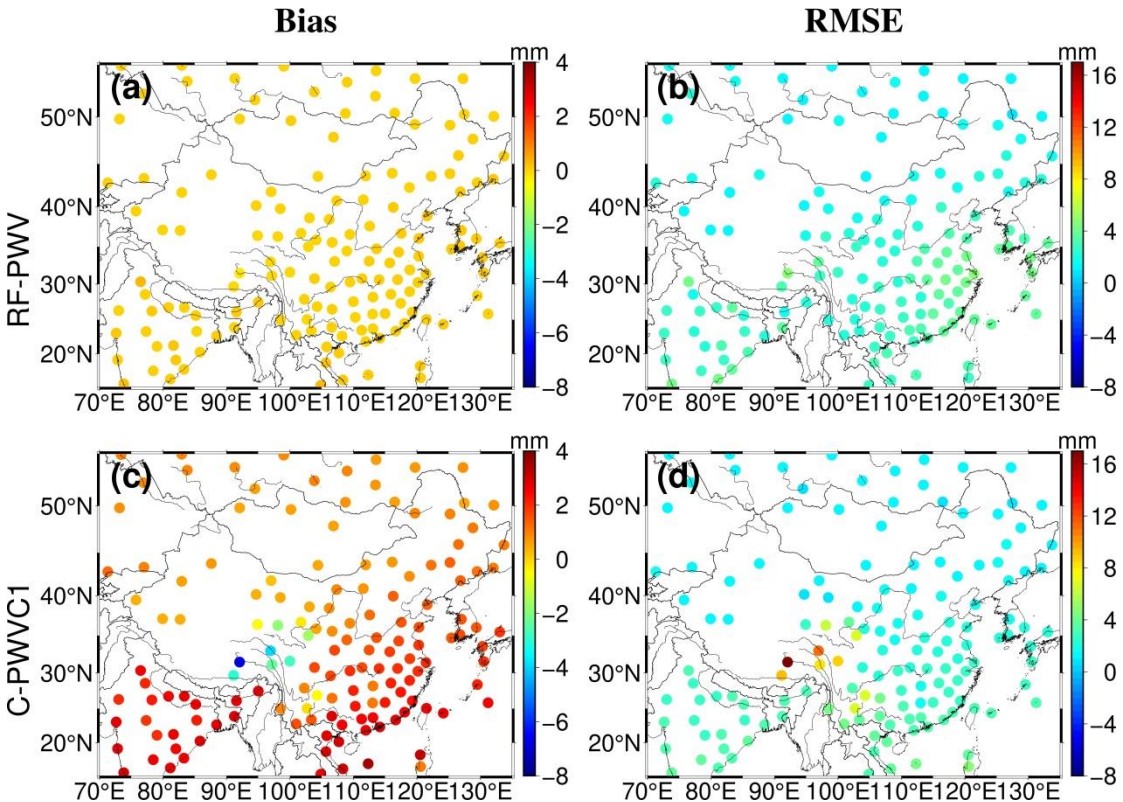

Figure 7. **Distributions of Bias and RMSE for the RF-PWV and C-PWVC1 with respect to the RS data**

The Bias and RMSE of RF-PWV and C-PWVC1 were also statistically analyzed for each month to assess the models' performance under different seasonal conditions. These results are presented in Figure 8. Notably, C-PWVC1 exhibits positive Bias in every month, indicating both systematic bias and clear seasonal variations. The Bias is minimal during winter (January, February, and December), with December showing the lowest value at 0.62 mm, while it reaches its peak during summer (June, July, and August), with a maximum value of 3.05 mm observed in August. In contrast, RF-PWV Bias demonstrates improvement in every month compared to C-PWVC1. Both models exhibit seasonal variation characteristics, with lower accuracy during summer and higher precision in winter. This seasonal variation is attributed to the warm and humid weather with abundant rainfall in summer, leading to significant PWV fluctuations. Nevertheless, RF-PWV still shows notable Bias optimization compared to C-PWVC1. Winters are typically drier and experience less rainfall, resulting in relatively smoother PWV changes. Consequently, both models can accurately capture PWV variations during this period, with RF-PWV having a distinct Bias advantage. Furthermore, the RMSE of RF-PWV and C-PWVC1 exhibits similar variations to Bias. While RF-PWV RMSE is slightly larger than that of C-PWVC1 in late autumn and winter, it is smaller than C-PWVC1 in other months, particularly during summer and early autumn. RF-PWV's advantage becomes more pronounced when dealing with spatio-temporal PWV variations that are more drastic. It is important to note that differences between validation results based on radiosonde and ERA5 data may be attributed to certain systematic deviations between





radiosonde and ERA5 data. In summary, RF-PWV demonstrates superior performance in vertical PWV correction under various seasonal conditions.

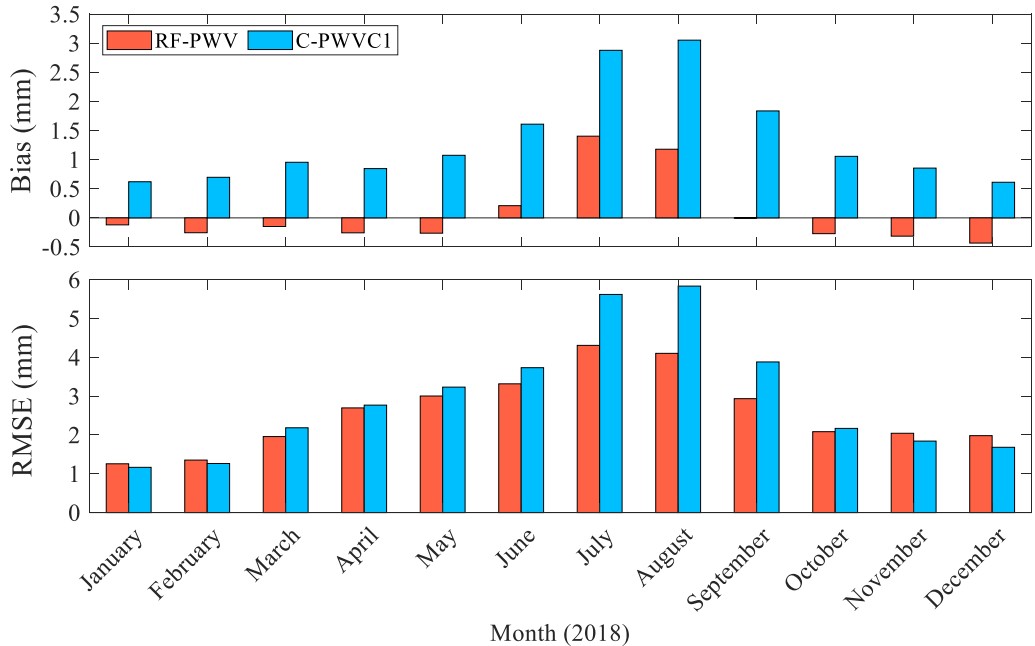

**Figure 8. The Bias and RMSE each month from the RF-PWV and C-PWVC1.**

290 To further evaluate the models' application in the vertical direction, the Bias and RMSE of RF-PWV and C-PWVC1 in different height difference segments were examined, and the results are depicted in Figure 9. In Figure 9a and c, for C-PWVC1, when the height difference is less than 0 km, the Bias and RMSE are –1.81 mm and 2.89 mm, respectively. As the height difference increases, both Bias and RMSE increase as well. When the height difference exceeds 2.5 km, the Bias stabilizes at 2 – 2.5 mm, while the RMSE remains around 3.5 mm. RF-PWV demonstrates higher accuracy and stability

295 across all height difference segments, with Bias approaching 0 mm and RMSE being smaller than that of C-PWVC1. Figure 9b and d depict the improvement rates of the absolute values of Bias and RMSE for RF-PWV compared to C-PWVC1 (Positive values indicate improvement). The absolute value of Bias exhibits an improvement rate of over 80%, with the maximum value approaching 100%. Meanwhile, the improvement rate of RMSE is significantly larger when the height difference is less than 3.5 km; it decreases slightly when the height difference exceeds 3.5 km but still remains around 5%. In

300 summary, RF-PWV offers higher vertical correction accuracy and improved stability across various height differences, demonstrating its strong applicability at different elevations.

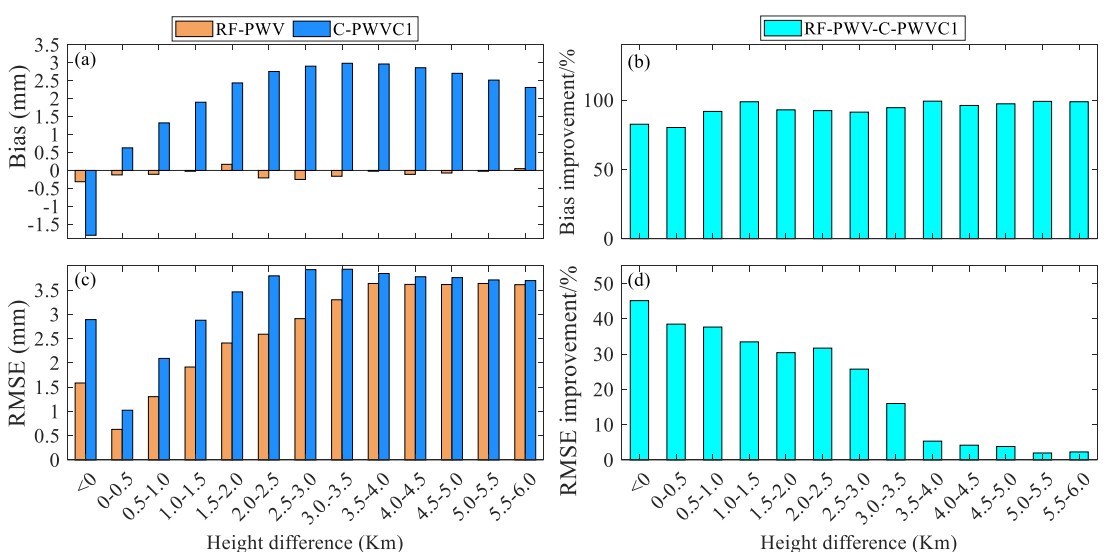

**Figure 9. Variation of Bias and RMSE with height differences (a, c) and improvement rates of the absolute values of Bias and RMSE (b, d).**

## 4 Conclusions and outlooks

Modeling accurate PWV vertical corrections benefits PWV fusion and provides detailed PWV vertical distribution information for meteorological studies. The complex terrain in China, characterized by varying climates and frequent water vapor exchanges, makes it challenging to accurately capture PWV variations at different heights. Consequently, this paper aims to develop a high-precision vertical PWV correction grid model. The primary contributions of this research can be summarized as follows:

(1). We establish a PWV vertically corrected grid model (RF-PWV) with a resolution of 1°×1° by integrating RF and monthly averaged hourly PWV data. This model utilizes RF to estimate the vertical variation of PWV at each grid point and demonstrates excellent applicability within a 6 km height difference. It effectively approximates PWV vertical changes. Validation against ERA5 data reveals that RF-PWV reduces Bias and RMSE by 99.84% and 63.40%, respectively, compared to C-PWVC1. RS validation also shows reductions of 96.36% in Bias and 5% in RMSE compared to C-PWVC1. Furthermore, RF-PWV exhibits robust resistance to seasonal and height differences interference.

(2). RF is employed to model each grid point (1°×1°), with the grid serving to decompose spatial variations and confine RF within the corresponding grid point. This simplifies the features of training samples for each grid point RF, potentially reducing the likelihood of RF getting stuck in a local optimum. Simultaneously, during training, issues with a particular grid will not impact the models of other grid points; thus, enhancing modeling efficiency. This approach also eliminates concerns about spatial generalization ability and ensures relatively stable accuracy across all grid points, contributing to the model's robustness.





Comprehensive validation demonstrates that RF-PWV can more accurately provide PWV vertical corrections in China and its surrounding areas. This model holds great potential for PWV vertical correction and is well-suited for delivering detailed

PWV vertical distribution information for multi-source water vapor fusion and meteorological research. Consequently, this method can be used to develop a globally applicable vertical correction model with higher accuracy, benefiting a wider range of users.

*Author contributions.* JL, YW, LL, YY, LH and FL Conceptualization, JL, YW and YY Methodology, JL and YW Formal

analysis, Writing-original draft, Writing review editing, JL, YW and LL Validation, JL, YW and YY Data curation, JL, LL and LH Funding acquisition, LL Investigation, YY Resources, YW and FL Software. All authors helped with discussions and with revising the manuscript.

*Data availability.* The radiosonde data are available on the website: https:/www.ncei.noaa.gov/pub/data/igra/. The ERA5

monthly averaged data on the following websites: https://cds.climate.copernicus.eu/

*Code availability.* The source code and model implementation used in this study are publicly available at https://github.com/jyli999/RF-PWV-model for interested readers to access and replicate the results presented in this paper. All of the data generated during the current study and the code are available on ZENODO

(https://zenodo.org/records/10124326).

*Competing interests.* The authors declare that they have no known competing financial interests or personal relationships that could have appeared to influence the work reported in this paper.

*Acknowledgments.* This work was supported by the National Natural Science Foundation of China (42304018), the Guangxi Natural Science Foundation of China (GuikeAD23026177, 2020GXNSFBA297145), the Foundation of Guilin University of Technology (GUTQDJJ6616032), Guangxi Key Laboratory of Spatial Information and Geomatics (21-238-21-05), the National Natural Science Foundation of China (42330105, 42064002, 42074035), and the Innovation Project of Guangxi Graduate Education (YCSW2023341).




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
