# Peer review of "A Grid Model for Vertical Correction of Precipitable Water Vapor over the Chinese Mainland and Surrounding Areas Using Random Forest"

_Geoscientific Model Development, 2023_

## Referee Comment (RC1)

Review of "A Grid Model for Vertical Correction of Precipitable Water Vapor over the Chinese Mainland and Surrounding Areas Using Random Forest"

**Summary**

This paper develops a machine learning based model, RF-PWV, which predicts the PWV difference between two height levels based on their height difference and the time information. RF-PWV is trained based on a 10-year-long ERA5 dataset. This study shows when only given the bottom-level ERA5-based PWV data, RF-PWV can well capture the time-dependent vertical distribution of PWV in ERA5, outperforming the earlier model C-PWVC1. When verified against the radiosondes profiles, RF-PWV also shows a marginal improvement in terms of RMSE compared to C-PWVC1.

The authors have effectively summarized the results from RF-PWV, and presented a comprehensive comparison between RF-PWV and C-PWVC1. While this paper is in general easy to follow, there are a few concerns regarding the RF-PWV model and its applicability, insufficient details of C-PWVC1, unclear motivations for comparing RF-PWV and C-PWVC1, and the ambiguities in the text and notations. Therefore, I recommend a major revision at this stage. I believe resolving these concerns can enhance the impact of this paper.

**Major comments**
1. About the RF-PWV model:
   (1) Why is 'year' included as an input variable to RF-PWV? Are there justifications for its relevance to the vertical distribution of PWV? Considering the model is trained on a 10-year dataset, which is relatively short, how reliable and generalizable is this relationship, even if one exists?

   (2) How is the 'day of the year' information obtained in the training data if the monthly averaged hourly dataset is used? My understanding is the monthly averaged hourly dataset for one year is a 24 (hr)*12(month) dataset, which should be something like:

   average PWV at 00:00 in January 2020
   average PWV at 01:00 in January 2020
   …
   average PWV at 23:00 in January 2020
   average PWV at 00:00 in February 2020

…

average PWV at 00:00 in December 2020

…

Is my understanding correct, or what am I missing? Please clarify this in the manuscript.

(3) I suggest expand on the practical applications of RF-PWV. For example, one could also use a data-assimilated product (like ERA5 and other reanalysis products) to obtain a vertical distribution of RF-PWV that has comparable accuracy with the RF-PWV output. Is there a current practical demand for precise real-time information of the vertical distribution of PWV that existing products are unable to provide? Please elaborate if applicable.

(4) Since RF-PWV is trained on ERA5, its accuracy is 'likely' (not necessarily, since ERA5 is not the truth) not superior than ERA5 itself when verified against accurate observations (e.g., radiosonde). Have you compared the error of RF-PWV and ERA5? Have you considered training a machine learning model purely based on accurate observations?

2.  What is C-PWVC1 model? A summary of how this model works is required. For example, what are the key properties that make RF-PWV better than C-PWVC1? How does C-PWVC1 use the time information in predicting the vertical distribution of PWV? Why do you choose to compare RF-PWV with C-PWVC1 (e.g., why not C-PWVC2)?

3.  There are a few ambiguities within the notations and nomenclature that might confuse and mislead readers. These ambiguities should be addressed and clarified to improve understanding. Please refer to the minor comments.

**Minor comments**
L13: PWV differences -> This is ambiguous. It should be something like the differences between …

L56: I suggest introduce the full name of RF-PWV here

L83-85: In the text, it is stated that "The PWV for each pressure level is determined …" while the PWV in Equation (1) is the integral of PWV of the whole column. Please

clarify the notations, e.g., using $PWV_i$ to denote the PWV for the i-th pressure level.

L125: 'test set' -> 'validation set'. I suggest only use test set to refer to the 2018 dataset.

L138-139: It would be helpful to use better notation here such as $\Delta PWV_i = PWV_i - PWV_0$, where 0 stands for the bottom level.

L140: "$22 \times 24 \times 12 \times 10$" Please be specific what each number refers to here. I am assuming 22 levels, 24 hours, 12 months, 10 years?

L143: "the reference PWV". The word 'reference' has also been used to refer to the 'true' dataset in the validation later in the paper (e.g., L164). It is somewhat confusing.

L140-145 & Figure 3: I suggest make it very clear of not only the input/output of the RF-PWV, but also the input/output variables of the ML part

L149: 'Subsequently, the target point'. This sentence needs to be rewritten. It's incomplete and unclear.

L160: What does 'C-PWVC1' stand for? A description of the C-PWVC1 model is missing (see major comment #2)

Figure 4&7: I suggest the colorbar for RMSE to only display positive values.

Figure 5: I suggest change the x-axis from "doy" to month

L238-248: I suggest elaborate in detail on how the validation is conducted in Section 3.2. It seems that the input to RF-PWV model is no longer the PWV difference relative to the bottom level (like what was done in Section 3.1). What are the input/output of RF-PWV, and what are the output verified against? Using a few simple equations with precise notations could be beneficial. Please clarify.

L292: What does it mean when the height difference is less than 0?

A general comment for Section 3: While the authors provide a thorough description of the plot/table results, I suggest trim some unnecessary details and focus more on explaining a few key findings and their implications, which can enhance the overall presentation.

---

## Author Comment (AC1)

The authors presented a PWV vertical correction grid model using the Random Forest and ERA5 monthly average hourly data. The performance of the new model was evaluated with ERA5 and radiosonde pressure levels PWV and the newly developed PWV vertical correction model (C-PWVC1) in aspects of seasonal and spatial locations and height differences. The results denoted that the new model is superior to C-PWVC1 across the research regions. Overall, the paper is written well and logically. Therefore, a minor revision is recommended for this manuscript before acceptance. However, to improve the quality of the paper, the following comments should be solved.

Response: Thanks for your valuable comments and suggestions on our manuscript, which are very helpful for improving our manuscript. We have carefully revised our manuscript, and the detailed revisions and responses are listed below:

1. L29-30: The discussion of "inconsistent pressure levels (heights) for storing PWV data from different sources hinder the fusion and reliability analysis of PWV multi-source data." is not convincing enough because lack of approved references.

Response: Sorry for the confusion. We have added references in L30-32 as follows: "However, inconsistent pressure levels (heights) for storing PWV data from different sources hinder the fusion and reliability analysis of PWV multi-source data (Chen et al., 2023b; Yang et al., 2023)".

References:

Chen, B. Y., Tan, J. S., Wang, W., Dai, W. J., Ao, M. S., and Chen, C. H.: Tomographic Reconstruction of Water Vapor Density Fields From the Integration of GNSS Observations and Fengyun-4A Products, IEEE Trans. Geosci. Remote Sensing, 61, 12, 10.1109/tgrs.2023.3239392, 2023b

Yang, F., Sun, Y. L., Meng, X. L., Guo, J. M., and Gong, X.: Assessment of tomographic window and sampling rate effects on GNSS water vapor tomography, Satell. Navig., 4, 12, 10.1186/s43020-023-00096-4, 2023.

2. Why Random Forest (RF) is employed for modeling, but not the commonly used Backpropagation (BP) neural network or other machine learning algorithms?

Response: Sorry for the confusion. We have added the reason for the Random Forest (RF) is employed for modeling. We have clarified this in L143-149 in the revised manuscript like: " The Random Forest, an ensemble learning method that combines multiple weak learners to form a single strong learner, typically improves generalization performance and model robustness. (Breiman, 2001;Sagi and Rokach, 2018). Compared to the Backpropagation neural network (BPNN), random forests are less prone to overfitting, especially with noisier datasets like PWV. Random forests handle noisy data and outliers more efficiently, making new models more robust and often easier to tune (Wang et al., 2016a;Tyralis et al., 2019). In addition, our previous study has shown that RF outperforms a popular algorithm of machine learning (BPNN) in modeling spatiotemporal variability in tropospheric parameters (Li et al., 2023). Thus, RF is employed to model the height dependency of PWV. "

References:

Breiman, L.: Random forests, Mach. Learn., 45, 5-32, 10.1023/a:1010933404324, 2001.

Sagi, O. and Rokach, L.: Ensemble learning: A survey, Wiley Interdiscip. Rev.-Data Mining Knowl. Discov., 8, 18, 10.1002/widm.1249, 2018.

Wang, L. A., Zhou, X. D., Zhu, X. K., Dong, Z. D., and Guo, W. S.: Estimation of biomass in wheat using random forest regression algorithm and remote sensing data, Crop J., 4, 212-219,

10.1016/j.cj.2016.01.008, 2016a

Tyralis, H., Papacharalampous, G., and Langousis, A.: A Brief Review of Random Forests for Water Scientists and Practitioners and Their Recent History in Water Resources, Water, 11, 37, 10.3390/w11050910, 2019.

Li, J. Y., Zhang, Q. L., Liu, L. L., Yao, Y. B., Huang, L. K., Chen, F. D., Zhou, L., and Zhang, B.: A refined zenith tropospheric delay model for Mainland China based on the global pressure and temperature 3 (GPT3) model and random forest, GPS Solut, 27, 13, 10.1007/s10291-023-01513-6, 2023.

3. L54: The first appearance of "RMSE" should explicitly refer to its full name. Additionally, please review all abbreviations to ensure they are defined at their first appearance (except in the abstract).

Response: Thank you for your reminder. We checked the manuscript and made the following modified in L77-78: " This model outperformed the Chinese Tropospheric Model (CTrop) and Global Pressure and Temperature 3 (GPT3), reducing Root Mean Square Error (RMSE) by 86% and 83%, respectively. "

4. L89: Please confirm the units for the explanation of $g$ in Eq (2).

Response: Thanks for your suggestion. We have modified it in L115-116 as follows: " $g$ is the gravitational acceleration (m/s$^2$), $\varphi$ denotes the latitude (rad). "

5. L143: What I am interested in is what are the respective 'day of the year (doy)' corresponding to the monthly average hourly dataset for each month when training the model. Please clarify these.

Response: Sorry for the confusion. We have analyzed the ERA5 monthly average hourly data and found that the "day of the year" is the first day of the corresponding month. We have clarified this in L192-194 in the revised manuscript like:" The input data included year, day of the year (doy is the first day of the corresponding month), hour of the day (hod), and $\Delta GPH$; the output data were $\Delta PWV$."

6. L149: 'Subsequently, the target point.'. Rephrase this sentence.

Response: Sorry for the confusion. According to your and another reviewer's comments, we have deleted this sentence in L211 to maintain a logical structure and improve the readability of this manuscript.

7. L193: We suggest that the color bar range for RMSE in Figure 4 should start from 0. The same is true of Figure 7.

Response: Thanks for your suggestion. According to your and another reviewer's comments, we have revised the color bar range for RMSE in Figure 4 and Figure 7, like: "

[Figure]

**Figure 4. Distributions of Bias and RMSE for the RF-PWV and C-PWVC1 with respect to the ERA5 data**

[Figure]

**Figure 7. Distributions of Bias and RMSE for the RF-PWV and C-PWVC1 with respect to the RS data"**

8.L244: What is the output of the four grid point model around the RS station? Please clarify it to help readers repeat your experiments.

Response: Sorry for the confusion. We have revised it in L352-378: " The datum PWV is the PWV corresponding to the surface height of the RS station. For each RS station, the four nearest grid points ($1°\times1°$) were selected, and the $\Delta PWV_i$ ($i$ =1,2,3,4, $i$ denotes the four nearest grid points) of the target height relative to the datum height computed based on the RF model of each grid point. Then $\Delta PWV_i$ were bilinearly interpolated to the corresponding location of the RS station to obtain the RF-PWV result. "

9. L257-259: The text about Figure 7b is not consistent with the fact.

Response: Thank you for your suggestion, this is our negligence. We checked the manuscript and made the following modified in L392-393: "As depicted in Figures 7a and 7c, C-PWVC1 exhibits a positive Bias on almost all stations except for the RS stations in the Yunnan-Guizhou Plateau, where the Bias is less pronounced. "

---

## Author Comment (AC2)

Reviewer 1: This paper develops a machine learning based model, RF-PWV, which predicts the PWV difference between two height levels based on their height difference and the time information. RF-PWV is trained based on a 10-year-long ERA5 dataset. This study shows when only given the bottom-level ERA5-based PWV data, RF-PWV can well capture the time-dependent vertical distribution of PWV in ERA5, outperforming the earlier model C-PWVC1. When verified against the radiosondes profiles, RF-PWV also shows a marginal improvement in terms of RMSE compared to C-PWVC1.

The authors have effectively summarized the results from RF-PWV, and presented a comprehensive comparison between RF-PWV and C-PWVC1. While this paper is in general easy to follow, there are a few concerns regarding the RF-PWV model and its applicability, insufficient details of C-PWVC1, unclear motivations for comparing RF-PWV and C-PWVC1, and the ambiguities in the text and notations. Therefore, I recommend a major revision at this stage. I believe resolving these concerns can enhance the impact of this paper.

Response: Thanks for your comments and suggestions. We have carefully revised the manuscript accordingly. Detailed revisions and responses are as follows.

**Major comments**

1. About the RF-PWV model:

(1) Why is 'year' included as an input variable to RF-PWV? Are there justifications for its relevance to the vertical distribution of PWV? Considering the model is trained on a 10-year dataset, which is relatively short, how reliable and generalizable is this relationship, even if one exists?

Response: Sorry for the confusion. The reason why 'year' included as an input variable to RF-PWV is that PWV lapse rate has a significant periodic function with year (Du et al., 2023; Huang et al., 2023). And what we did is just follow the conventional method as presented by Bohm et al.,2015, and Landskron and Bohm,2018, who used year-related Julian days as input variable to model the vertical dependence of tropospheric parameters. The most widely used models (Bohm et al.,2015; Landskron and Bohm,2018) are now also based on 10 years of data, to which the addition of hourly-resolution data in this paper is expected to yield more accurate height correction factors. We have clarified this in L194-195 in the revised manuscript like: "The reason why 'year' included as an input variable to RF-PWV is that PWV lapse rate has a significant periodic function with year (Du et al., 2023; Huang et al., 2023). "

References:

Böhm, J., Möller, G., Schindelegger, M., Pain, G., and Weber, R.: Development of an improved empirical model for slant delays in the troposphere (GPT2w), GPS Solut, 19, 433-441, 10.1007/s10291-014-0403-7, 2015.

Landskron, D. and Böhm, J.: VMF3/GPT3: refined discrete and empirical troposphere mapping functions, J. Geodesy, 92, 349-360, 10.1007/s00190-017-1066-2, 2018.

Huang, L. K., Liu, W., Mo, Z. X., Zhang, H. X., Li, J. Y., Chen, F. D., Liu, L. L., and Jiang, W. P.: A new model for vertical adjustment of precipitable water vapor with consideration of the time-varying lapse rate, GPS Solut, 27, 16, 10.1007/s10291-023-01506-5, 2023

Du, Z., Yao, Y. B., and Zhao, Q. Z.: Novel Validation and Calibration Strategy for Total Precipitable Water Products of Fengyun-2 Geostationary Satellites, IEEE Trans. Geosci. Remote Sensing, 61, 12, 10.1109/tgrs.2023.3295091, 2023.

(2) How is the 'day of the year' information obtained in the training data if the monthly averaged hourly dataset is used? My understanding is the monthly averaged hourly dataset for one year is a 24 (hr)*12(month) dataset, which should be something like:

average PWV at 00:00 in January 2020

average PWV at 01:00 in January 2020

...

average PWV at 23:00 in January 2020

average PWV at 00:00 in February 2020

...

average PWV at 00:00 in December 2020

...

Is my understanding correct, or what am I missing? Please clarify this in the manuscript.

Response: Sorry for the confusion. Your understanding is right, we have analyzed the ERA5 monthly average hourly data and found that the "day of the year" is the first day of the corresponding month. We have clarified this in L193 in the revised manuscript like:" The input data included year, day of the year (doy is the first day of the corresponding month), hour of the day (hod), and $\Delta GPH$; the output data were $\Delta PWV$."

(3) I suggest expand on the practical applications of RF-PWV. For example, one could also use a data-assimilated product (like ERA5 and other reanalysis products) to obtain a vertical distribution of RF-PWV that has comparable accuracy with the RF-PWV output. Is there a current practical demand for precise real-time information of the vertical distribution of PWV that existing products are unable to provide? Please elaborate if applicable.

Response: Thanks for your suggestion. We have expanded on the practical applications of PWV vertical distribution and have modified it in L42-46 as follows: "The vertical distribution of PWV is closely related to the formation and distribution of rainfall and clouds, which is of great help to weather forecasting and is also one of the factors affecting convection and monsoon climates(Bevis et al., 1992;Keil et al., 2008;Rose and Rencurrel, 2016). The vertical distribution of PWV and its temporal variability is essential for understanding regional weather and global climate, improving the climate models, and predicting future climate change(Jacob, 2001;Renju et al., 2015) "

References:

Bevis, M., Businger, S., Herring, T. A., Rocken, C., Anthes, R. A., and Ware, R. H.: GPS METEOROLOGY - REMOTE-SENSING OF ATMOSPHERIC WATER-VAPOR USING THE GLOBAL POSITIONING SYSTEM, J. Geophys. Res.-Atmos., 97, 15787-15801, 10.1029/92jd01517, 1992.

Keil, C., Röpnack, A., Craig, G. C., and Schumann, U.: Sensitivity of quantitative precipitation forecast to height dependent changes in humidity, Geophys. Res. Lett., 35, 5, 10.1029/2008gl033657, 2008.

Rose, B. E. J. and Rencurrel, M. C.: The Vertical Structure of Tropospheric Water Vapor: Comparing Radiative and Ocean-Driven Climate Changes, J. Clim., 29, 4251-4268, 10.1175/jcli-d-15-0482.1, 2016.

Jacob, D.: The role of water vapour in the atmosphere. A short overview from a climate modeller's point of view, Phys. Chem. Earth Pt. A-Solid Earth Geod., 26, 523-527, 10.1016/s1464-1895(01)00094-1, 2001.

Renju, R., Raju, C. S., Mathew, N., Antony, T., and Moorthy, K. K.: Microwave radiometer observations of interannual water vapor variability and vertical structure over a tropical station, J. Geophys. Res.-Atmos., 120, 4585-4599, 10.1002/2014jd022838, 2015

(4) Since RF-PWV is trained on ERA5, its accuracy is 'likely' (not necessarily, since ERA5 is not the truth) not superior than ERA5 itself when verified against accurate observations (e.g., radiosonde). Have you compared the error of RF-PWV and ERA5? Have you considered training a machine learning model purely based on accurate observations?

Response: Thanks for the reviewer's suggestion. We actually did what the reviewer suggested in Section 3.1 (use not involved in modeling ERA5 data to validate the model). We agree with your statement that the results of validation with RS data are indeed slightly weaker than the results of validation with ERA5. We have clarified this in L388-390 in the revised manuscript like: "Moreover, these results show that the accuracy analyzed by RS data is slightly lower to those estimated by ERA5 data. This is because of the significant systematic bias between ERA5 and RS(Zhu et al., 2022; Sun et al., 2019) but such accuracy can still meet the meteorological requirements for PWV accuracy". Moreover, we use the ERA5 grid data to construct the model at each grid node, one of the purposes of which is to reduce the excessive loss of accuracy that occurs when the target station is at a large distance from the modeled station. Modeling using only accurate observations (e.g., radiosonde) has the possibility of obtaining higher accuracy corrections effect, but the spatial resolution of the RS data in the study area is low and the distribution of the RS stations in the western part of the study area is relatively sparse. If modeling is based only on RS data, severe local loss of accuracy may occur(Gui et al., 2017; Zhu et al., 2021). Therefore, we did not consider modeling directly based on accurate RS data.

References:

Zhu, M. C., Yu, X. W., and Sun, W.: A coalescent grid model of weighted mean temperature for China region based on feedforward neural network algorithm, GPS Solut, 26, 11, 10.1007/s10291-022-01254-y, 2022.

Sun, Z. Y., Zhang, B., and Yao, Y. B.: An ERA5-Based Model for Estimating Tropospheric Delay and Weighted Mean Temperature Over China With Improved Spatiotemporal Resolutions, Earth Space Sci., 6, 1926-1941, 10.1029/2019ea000701, 2019.

Gui, K., Che, H. Z., Chen, Q. L., Zeng, Z. L., Liu, H. Z., Wang, Y. Q., Zheng, Y., Sun, T. Z., Liao, T. T., Wang, H., and Zhang, X. Y.: Evaluation of radiosonde, MODIS-NIR-Clear, and AERONET precipitable water vapor using IGS ground-based GPS measurements over China, Atmos. Res., 197, 461-473, 10.1016/j.atmosres.2017.07.021, 2017.

Zhu, H., Chen, K. J., and Huang, G. W.: A Weighted Mean Temperature Model with Nonlinear Elevation Correction Using China as an Example, Remote Sens., 13, 13, 10.3390/rs13193887, 2021.

2. What is C-PWVC1 model? A summary of how this model works is required. For example, what are the key properties that make RF-PWV better than C-PWVC1? How does C-PWVC1 use the time information in predicting the vertical distribution of PWV? Why do you choose to compare RF-PWV with C-PWVC1 (e.g., why not CPWVC2)?

Response: Thanks for your suggestion. We have furtherly clarified C-PWVC1 model and key properties that make RF-PWV better than C-PWVC1 in L233-243 in the revised manuscript, like: " To validate the RF-PWV model, we employed hourly ERA5 and RS pressure level data from the study area in 2018 as the test set, while also selecting a newly developed PWV vertical correction

model (C-PWVC1) for comparison. Note that the authors of the C-PWVC model suggest using C-PWVC1 directly for PWV vertical correction in the study area, so C-PWVC2 is ignored. C-PWVC1 has been proven to be more accurate than the classical PWV vertical correction model (PWV lapse rate = –0.5 mm/km) in the study area(Huang et al., 2021). C-PWVC1 is a model using the exponential function to account for the height dependency of PWV. C-PWVC1 can be expressed as follows:

$$PWV_{h_1} = PWV_{h_2} \bullet \exp\big(\beta(h_1 - h_2)\big), \tag{8}$$

$$\beta(doy) = -0.35 - 0.026\cos\left(\frac{doy}{365.25}2\pi\right) - 0.015\sin\left(\frac{doy}{365.25}2\pi\right) + 0.008\cos\left(\frac{doy}{365.25}4\pi\right) +$$

$$0.026\sin\left(\frac{doy}{365.25}4\pi\right), \tag{9}$$

where $PWV_{h_1}$ and $PWV_{h_2}$ denote the PWV at $h_1$ and $h_2$ respectively, $\beta$ is the PWV lapse rate, and $doy$ is the day of the year. C-PWVC1 requires inputs of datum height, datum PWV, target height, and time to provide the PWV correction value at the target height, but the model is unable to capture nonlinear variations in the vertical direction."

The accuracy of C-PWVC2 is lower than that of C-PWVC1 in other places except the Tibetan Plateau, and the overall accuracy is similar to C-PWVC1; on the other hand, the authors of the C-PWVC model suggest using C-PWVC1 directly for PWV vertical correction in the study area. Therefore, C-PWVC1 is used directly in the study. We have clarified this in L234-236 in the revised manuscript like: "Note that the authors of the C-PWVC model suggest using C-PWVC1 directly for PWV vertical correction in the study area, so C-PWVC2 is ignored. "

References:

Huang, L., Mo, Z., Liu, L., and Xie, S.: An empirical model for the vertical correction of precipitable water vapor considering the time-varying lapse rate for Mainland China, Acta Geodaetica et Cartographica Sinica, 50, 1320-1330, 10.11947/j.AGCS.2021.20200530, 2021.

3. There are a few ambiguities within the notations and nomenclature that might confuse and mislead readers. These ambiguities should be addressed and clarified to improve understanding. Please refer to the minor comments.

Response: Thanks for your valuable comments and suggestions on our manuscript. We have revised the manuscript carefully according to your comments as follows:

**Minor comments**

L13: PWV differences -> This is ambiguous. It should be something like the differences between …

Response: Sorry for the confusion. We added the relevant notation to avoid confusion for readers, modifying L11-15 to the following: "Our model, known as RF-PWV (a PWV vertical correction grid model with a 1°x 1°resolution), is constructed using random forest based on the relationship between the differences from different pressure level PWV from the fifth-generation European Centre for Medium-Range Weather Forecasts reanalysis (ERA5) monthly average hourly data and corresponding differences from their heights differences over time. "

L56: I suggest introduce the full name of RF-PWV here

Response: Thanks for your suggestion. We have modified it in L69-71 as follows: "Therefore, this paper presents a Random Forest-based Precipitable water vapor vertical correction grid model, termed RF-PWV, for China and surrounding areas, harnessing Random Forest's powerful

nonlinear fitting capability and the high temporal resolution of monthly average hourly PWV data.
"

L83-85: In the text, it is stated that "The PWV for each pressure level is determined …" while the PWV in Equation (1) is the integral of PWV of the whole column. Please clarify the notations, e.g., using $PWV_i$ to denote the PWV for the i-th pressure level.

Response: Thanks for your suggestion. We have clarified Equation (1).as follows:"

$$PWV_i = \sum_i^{n-1} \frac{(q_i+q_{i+1})\bullet(p_{i+1}-p_i)}{2\bullet\rho_w\bullet g}, \tag{1}$$

where $n$ represents the total number of layers, $PWV_i$, $q_i$ and $p_i$ represent the PWV (mm), specific humidity (kg/kg) and pressure (Pa) at the $i$ layer, respectively; "

L125: 'test set' -> 'validation set'. I suggest only use test set to refer to the 2018 dataset.

Response: Thanks for your suggestion. We have revised the "test set" to "validation set" in L175 and clarified in L233-234 that the 2018 data is the test set.

L138-139: It would be helpful to use better notation here such as $\Delta PWV_i = PWV_i - PWV_0$, where 0 stands for the bottom level

Response: Thanks for your suggestion. We have modified L188-190, like:" The i-th PWV differences ($\Delta PWV_i = PWV_i - PWV_0$, where 0 stands for the bottom level) between $i$ level and bottom level and the responding height differences ($\Delta GPH_i = GPH_i - GPH_0$) were all computed and utilized as the training dataset."

L140: "22 × 24 × 12 × 10" Please be specific what each number refers to here. I am assuming 22 levels, 24 hours, 12 months, 10 years?

Response: Thanks for your suggestion. We have modified it in L190-191 as follows: "In essence, each grid point contained 63,360 samples (22 levels×24 hours×12 months×10 years), and the region consisted of 2,706 grid points (66 longitudes ×41 latitudes) at 1°×1°resolution. "

L143: "the reference PWV". The word 'reference' has also been used to refer to the 'true' dataset in the validation later in the paper (e.g., L164). It is somewhat confusing.

Response: Sorry for the confusion. We have revised it in L195-207: " When users employ the model, they are only required to provide the geopotential height of the target point, the datum PWV, the time (year, doy, hod), and the height difference of the target point concerning the datum point ($\Delta GPH$). They can obtain the corresponding $\Delta PWV$. And then, they can get the PWV of the target height by adding the datum PWV to the $\Delta PWV$. "

L140-145 & Figure 3: I suggest make it very clear of not only the input/output of the RF-PWV, but also the input/output variables of the ML part

Response: Thanks for your suggestion. We have added the ML part input/output in Figure 3.

[Figure]

Figure 3. Network structure of RF-PWV model based on random forest algorithm

L149: 'Subsequently, the target point'. This sentence needs to be rewritten. It's incomplete and unclear.

Response: Sorry for the confusion. According to your and another reviewer's comments, we have deleted this sentence in L211 to maintain a logical structure and improve the readability of this manuscript.

L160: What does 'C-PWVC1' stand for? A description of the C-PWVC1 model is missing (see major comment #2)

Response: Thanks for your suggestion. The revisions and responses are the same as in major comment #2.

Figure 4&7: I suggest the colorbar for RMSE to only display positive values

Response: Thanks for your suggestion. We have revised the color bar range for RMSE in Figure 4 and Figure 7 to only display positive values, like "

[Figure]

Figure 4. Distributions of Bias and RMSE for the RF-PWV and C-PWVC1 with respect to the ERA5 data

[Figure]

Figure 7. Distributions of Bias and RMSE for the RF-PWV and C-PWVC1 with respect to the RS data"

Figure 5: I suggest change the x-axis from "doy" to month

Response: Thanks for your suggestion. We modified the X-axis of Figure 5 from "doy" to month.

[Figure]

Figure 5. Time series of RF-PWV and C-PWVC1 Bias and RMSE on four selected grid points

L238-248: I suggest elaborate in detail on how the validation is conducted in Section 3.2. It seems that the input to RF-PWV model is no longer the PWV difference relative to the bottom level (like what was done in Section 3.1). What are the input/output of RF-PWV, and what are the output verified against? Using a few simple equations with precise notations could be beneficial. Please clarify.

Response: Thanks for your suggestion. The RF-PWV does not have as input the PWV difference relative to the bottom level, the input is the height difference of the target height relative to the height corresponding to the datum PWV, and the time (Year, doy, hod). The height corresponding to this datum PWV in section 3.2 is the surface height corresponding to the RS station. This has been clarified in L352-378, like: " The datum PWV is the PWV corresponding to the surface height of the RS station. For each RS station, the four nearest grid points $(1° × 1°)$ were selected, and the $\Delta PWV_i$ ($i$ =1,2,3,4, $i$ denotes the four nearest grid points) of the target height relative to the datum height computed based on the RF model of each grid point. Then $\Delta PWV_i$ were bilinearly interpolated to the corresponding location of the RS station to obtain the RF-PWV result "

L292: What does it mean when the height difference is less than 0?

Response: Sorry for the confusion. The occurrence of height differences less than 0 is due to the fact that in order to obtain a uniform distributed PWV in the height direction. This has been clalified in L348-353 and L470-473, like: " Since the stratified RS data are not uniformly distributed vertically, the variation of PWV with elevation was fitted using an exponential function based on the 2018 PWV data from each RS station. Using the fitting results, the PWVs of neighboring levels were interpolated using inverse distance weighting (IDW) to generate a sequence of PWVs within the range of 0–12 km with intervals of 500 m. This sequence of PWVs served as reference values. The datum PWV is the PWV corresponding to the surface height of the RS station. " and " It is noted that in order to obtain PWVs that are uniformly distributed in the height direction, we obtained PWVs with heights in the range of 0-12 KM; when the surface heights of some RS stations are greater than 0, their height differences relative to height 0 are less than 0. "

A general comment for Section 3: While the authors provide a thorough description of the plot/table results, I suggest trim some unnecessary details and focus more on explaining a few key findings and their implications, which can enhance the overall presentation.

Response: Thanks for your suggestion. We have deleted some most unnecessary sentences and focused on explaining a few key findings and their implications further, like: "  ","  ","  ","  ","  ","  ","  ","  ", "  ","  ".